# Screener: Self-supervised Pathology Segmentation Model for 3D Medical Images

## Abstract

Accurate segmentation of all pathological findings in 3D medical images remains a significant challenge, as supervised models are limited to detecting only the few pathology classes annotated in existing datasets. To address this, we frame pathology segmentation as an unsupervised visual anomaly segmentation (UVAS) problem, leveraging the inherent rarity of pathological patterns compared to healthy ones. We enhance the existing density-based UVAS framework with two key innovations: (1) dense self-supervised learning (SSL) for feature extraction, eliminating the need for supervised pre-training, and (2) learned, masking-invariant dense features as conditioning variables, replacing hand-crafted positional encodings. Trained on over 30,000 unlabeled 3D CT volumes, our model, Screener, outperforms existing UVAS methods on four large-scale test datasets comprising 1,820 scans with diverse pathologies. Code and pre-trained models will be made publicly available.

## 1. Introduction

Accurate identification, localization, and classification of *all* pathological findings in 3D medical images remain a significant challenge in medical computer vision. While supervised models have shown promise, their utility is limited by the scarcity of labeled datasets, which often contain annotations for only a few pathologies. For example, Figure 1 shows 2D slices of 3D computed tomography (CT) images (first row) from public datasets (Armato III et al., 2011; Tsai et al., 2020; Heller et al., 2019; Bilic et al., 2023) providing annotations of lung cancer, kidney tumors or liver tumors, while annotations of other pathologies, e.g., pneumothorax, are missing. This restricts the functionality of supervised models to narrow, task-specific applications.

[1] Anonymous Institution, Anonymous City, Anonymous Region, Anonymous Country. Correspondence to: Anonymous Author <anon.email@domain.com>.

Preliminary work. Under review by the International Conference on Machine Learning (ICML). Do not distribute.

Unlabeled CT images, however, are abundant: large-scale datasets (Team, 2011; Ji et al., 2022; Qu et al., 2024) are publicly available but often remain unused for training. Leveraging these datasets, we aim to develop an unsupervised model capable of distinguishing pathological regions from normal ones. Our core assumption is that pathological patterns are significantly rarer than healthy patterns in CT images. This motivates framing pathology segmentation as an unsupervised visual anomaly segmentation (UVAS) problem, where anomalies correspond to pathological regions.

While existing UVAS methods have been explored extensively for natural images, their adaptation to medical imaging is challenging. One obstacle is that uncurated CT datasets include many patients with pathologies, and there is no automatic way to filter them out to ensure a training set composed entirely of normal (healthy) images — a common requirement for synthetic-based (Zavrtanik et al., 2021; Marimont & Tarroni, 2023) and reconstruction-based (Baur et al., 2021; Schlegl et al., 2019) UVAS methods.

Density-based approaches are better suited for this setting because they model the distribution of image patterns probabilistically and assume that abnormal patterns are rare rather than entirely absent in the training dataset. To model the density of image patterns, these methods encode them into vector representations using a pre-trained encoder. The existing methods (Gudovskiy et al., 2022; Zhou et al., 2024) rely on encoders pre-trained on ImageNet (Deng et al., 2009), and their performance degrades when applied to medical images due to the significant domain shift. One could using medical domain-specific supervised encoders, such as STU-Net (Huang et al., 2023). However, our experiments show that this approach also works poorly, likely because the features learned by supervised encoders are too specific and do not contain information needed for distinguishing between pathological and healthy image regions.

To address these challenges, we propose using dense self-supervised learning (SSL) methods (O. Pinheiro et al., 2020; Wang et al., 2021; Bardes et al., 2022; Goncharov et al., 2023) to pre-train informative feature maps of CT images and employ them in the density-based UVAS framework. Thus, our model learns the distribution of dense SSL embeddings and assigns high anomaly scores to image regions where embeddings fall into low-density regions.

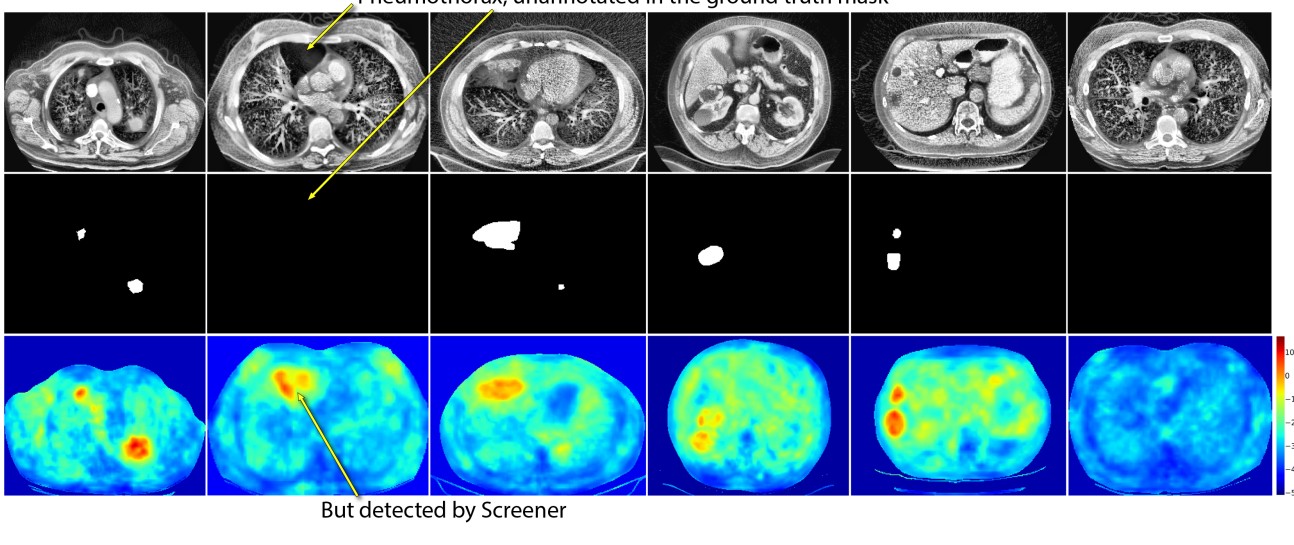

*Figure 1.* Examples of 2D slices of 3D medical CT images (the first row), the ground truth masks of their pathological regions (the second row) and the anomaly maps predicted by our fully self-supervised Screener model (the third row). Note that, the second image from the left contains pneumothorax, missed by ground truth annotation mask, but detected by our model.

Inspired by dense self-supervised learning, we also generalize the idea of conditioning in density-based UVAS methods. Existing works (Gudovskiy et al., 2022; Zhou et al., 2024) use hand-crafted conditioning variables like standard positional embeddings. We propose to replace them by pre-trained dense self-supervised features capturing context, i.e. global characteristics, of individual image regions, e.g. their anatomical position, patient's age. At the same time, we eliminate local information about presence of pathologies from the learned conditioning variables by enforcing their invariance to image masking.

We refer to the resulting model as Screener and train it on over 30,000 unlabeled CT volumes spanning chest and abdominal regions. As shown in Figure 1 (third row), our model successfully segments pathological regions across different organs. We demonstrate the Screener's superior performance compared to baseline UVAS methods on four large-scale test datasets comprising 1,820 scans with diverse pathologies. As shown in Figure 1, Screener, being a fully unsupervised model, demonstrates remarkable performance across diverse organs and conditions.

Our key contributions are three-fold:

- **Self-supervised encoder in density-based UVAS.** We demonstrate that dense self-supervised representations can be successfully used and even preferred over supervised feature extractors in density-based UVAS methods. This enables a novel fully self-supervised UVAS framework applicable in domains with limited labeled data.

- **Learned conditioning variables.** We introduce novel self-supervised conditioning variables for density-based models, simplifying the estimation of conditional distributions and achieving remarkable segmentation performance using a simple Gaussian density model.

- **First large-scale study of UVAS in CT images.** This work presents the first large-scale evaluation of UVAS methods for CT images, showing state-of-the-art performance on unsupervised semantic segmentation of pathologies in diverse anatomical regions, including lung cancer, pneumonia, liver and kidney tumors.

## 2. Background & notation

### 2.1. Density-based UVAS

The core idea of density-based UVAS methods is to assign high anomaly scores to image regions containing rare patterns. To implement this idea they involve two models, which we call a *descriptor model* and a *density model*. The descriptor model encodes image patterns into vector representations, while the density model learns their distribution and assigns anomaly scores based on the learned density.

The descriptor model $f_{\theta^{\text{desc}}}$ is usually a pre-trained fully-convolutional neural network. For a 3D image $\mathbf{x} \in \mathbb{R}^{H \times W \times S}$, it produces feature maps $\mathbf{y} \in \mathbb{R}^{h \times w \times s \times d^{\text{desc}}}$, where each position $p \in P$ corresponds to a descriptor $\mathbf{y}[p] \in \mathbb{R}^{d^{\text{desc}}}$. Here, position set $P = \{p \mid p \in [1, \ldots, h] \times [1, \ldots, w] \times [1, \ldots, s]\}$.

The density model $q_{\theta^{\text{dens}}}(y)$ estimates the marginal density

$q_Y(y)$ of descriptors ($Y$ denotes the descriptor at a random position in a random image). For an abnormal pattern at position $p$, the descriptor $\mathbf{y}[p]$ is expected to lie in a low-density region, yielding a low $q_{\theta^{\mathrm{dens}}}(\mathbf{y}[p])$. Conversely, normal patterns correspond to high density values. During inference, the negative log-density values, $-\log q_{\theta^{\mathrm{dens}}}(\mathbf{y}[p])$ are used as anomaly segmentation scores.

This framework can be extended using a conditioning mechanism. For each position $p$, one can introduce an auxiliary variable $\mathbf{c}[p]$, referred to as a *condition*. Then, instead of modeling the complex marginal density $q_Y(y)$, the conditional density $q_{Y|C}(y \mid c)$ is learned for each condition $c$ ($C$ denotes the condition at a random position in a random image). At inference, the negative log-conditional densities, $-\log q_{\theta^{\mathrm{dens}}}(\mathbf{y}[p] \mid \mathbf{c}[p])$, are used as anomaly scores. State-of-the-art methods (Gudovskiy et al., 2022; Zhou et al., 2024) adopt this conditional framework and use sinusoidal positional encodings as conditions.

### 2.2. Dense joint embedding SSL

Joint embedding self-supervised learning (SSL) methods learn meaningful image representations without labeled data by generating positive pairs—multiple views of the same image created through augmentations like random crops and color jitter. These methods learn embeddings that capture mutual information between views, ensuring they are informative (discriminating between images) and invariant to augmentations (predictable across views). Contrastive methods, e.g., SimCLR (Chen et al., 2020), explicitly push apart embeddings of different images, while non-contrastive methods, e.g., VICReg (Bardes et al., 2021), avoid degenerate solutions through regularization. Details on SimCLR and VICReg objectives are in the Appendix A.

*Dense* joint embedding SSL methods extend this idea by learning dense feature maps—pixel-wise embeddings that encode information about different spatial locations in an image. Instead of treating the entire image as a single entity, these methods define positive pairs at the pixel level: two embeddings form a positive pair if they correspond to the same absolute position in the original image but are predicted from different augmented crops. During training, dense SSL enforces similarity between positive pairs while avoiding collapse by encouraging dissimilarity between embeddings from different images or positions. DenseCL (Wang et al., 2021) and VADER (O. Pinheiro et al., 2020) use contrastive objectives, while VICRegL (Bardes et al., 2022) adopts a non-contrastive approach, regularizing the covariance matrix of embeddings to increase informational content. These methods excel at capturing fine-grained spatial information, making them ideal for tasks like object detection and segmentation.

## 3. Method

Our method introduces two key innovations to the density-based UVAS framework, described in Section 2.1: *self-supervised descriptor model*, and *self-supervised condition model*. The following Sections 3.1 and 3.2 describe these modules, while Section 3.3 describes details of density modeling. Figure 2 illustrates the overall training pipeline.

### 3.1. Descriptor model

The descriptor model plays a crucial role in our method. It must generate descriptors that effectively differentiate between pathological and normal positions; otherwise, these positions cannot be assigned distinct anomaly scores within the density-based UVAS framework. At the same time, the descriptors should minimize the inclusion of irrelevant information. For instance, if the descriptors capture noise – a common artifact in CT images – the density model may assign high anomaly scores to healthy regions with extreme noise values, leading to false positive errors.

To pre-train the descriptor model, we use dense joint embedding SSL methods described in Section 2.2, which allow explicit control over the information content of the representations. Specifically, we penalize descriptors for failing to distinguish between different positions within or across images, ensuring they capture spatially discriminative features. Simultaneously, we enforce invariance to low-level perturbations, such as cropping and color jitter, to eliminate irrelevant information.

The descriptor model training pipeline is illustrated in the upper part of Figure 2. From a random CT volume $\mathbf{x}$, we extract two overlapping 3D crops of random size, resize them to $H \times W \times S$, and apply random augmentations, such as color jitter. The augmented crops, denoted as $\mathbf{x}^{(1)}$ and $\mathbf{x}^{(2)}$, are fed into the descriptor model, producing feature maps $\mathbf{y}^{(1)}$ and $\mathbf{y}^{(2)}$.

From the overlapping region of the two crops, we randomly select $n$ positions. For each position $p$, we compute its coordinates $p^{(1)}$ and $p^{(2)}$ relative to the augmented crops and extract descriptors $y^{(1)} = \mathbf{y}^{(1)}[p^{(1)}]$ and $y^{(2)} = \mathbf{y}^{(2)}[p^{(2)}]$. These descriptors form a *positive pair*, as they correspond to the same position in the original image but are predicted from different augmentations.

Repeating this process for $m$ different seed CT volumes yields a batch of $N = n \cdot m$ positive pairs, denoted as $\{(y_i^{(1)}, y_i^{(2)})\}_{i=1}^N$. Given this batch, we optimize the descriptor model with standard SSL objectives: InfoNCE (Chen et al., 2020) or VICReg (Bardes et al., 2021), detailed in Appendix A.

Conceptually, our descriptor model is similar to dense SSL models described in Section 2.2. However, our implementa-

*Figure 2.* Illustration of Screener. First, we pre-train a self-supervised descriptor model to produce informative feature maps which are invariant to image crops and color jitter. Second, we train a self-supervised condition model in the same way as the descriptor model, but also enforcing invariance to masking of random image blocks. Thus, condition model feature maps are ignorant about anomalies and contain only the information that can be always inferred from the unmasked context. Third, density model learns the conditional distribution $p_{Y|C}(y \mid c)$ of feature vectors $Y = y[p]$ and $C = c[p]$ produced by descriptor and condition models at random image position $p$. To obtain a map of anomaly scores we apply density model in a pixel-wise manner, which can be efficiently implemented using $1 \times 1 \times 1$ convolutions.

tion have many important differences. In contrast to (Wang et al., 2021; O. Pinheiro et al., 2020; Bardes et al., 2022), our model has a UNet-like architecture and its output feature maps have very high resolution ($h \times w \times s = H \times W \times S$), which is a common standard for 3D medical image segmentation. (Wang et al., 2021; O. Pinheiro et al., 2020) do not

treat embeddings from the same image as negatives as we do. We do not employ any auxiliary global SSL objectives, like (Wang et al., 2021; Bardes et al., 2022). And we do not obtain position-wise descriptors by concatenating features from feature pyramid, as in (Goncharov et al., 2023). Other implementation details are described in Appendix D.

### 3.2. Condition model

Our self-supervised condition model is inspired by a thought experiment: imagine a region of a CT image is masked, and we attempt to infer its content based on the visible context (see masked crops in Figure 2 for illustration). In most cases, we would assume the masked region is healthy unless there is explicit evidence suggesting otherwise. This assumption reflects our model of the conditional distribution over possible inpaintings given the context. If the actual content deviates significantly from this distribution, we treat it as an anomaly.

This intuition suggests that the condition $\mathbf{c}[p]$ in the conditional density-based UVAS framework should capture the *global* context of the image position $p$. *Global* implies that $\mathbf{c}[p]$ must be inferable from various masked views of the image. At the same time, conditions should vary across different images and regions within the same image to encode position-specific or patient-specific information effectively.

To achieve these properties, we propose learning conditions $\mathbf{c}[p]$ using a self-supervised condition model $g_{\theta^{\mathrm{cond}}}$. This model shares the same fully convolutional architecture as the descriptor model and produces conditions $\{\mathbf{c}[p]\}_{p \in P}$ in the form of feature maps $\mathbf{c} \in \mathbb{R}^{h \times w \times s \times d^{\mathrm{cond}}}$. To ensure conditions are inferable from any masked image view, we enforce feature maps invariance with respect to random image masking during training. Thus, the training procedure mirrors the training of the descriptor model (Section 3.1), with masking incorporated as part of the augmentations. An illustration of this approach is shown in the middle part of Figure 2.

The learned conditions $\mathbf{c}[p]$ are designed to ignore the presence of pathologies, as such information cannot be consistently inferred from masked views. Instead, the condition model likely encodes patient-level attributes (e.g., age, gender) and position-specific attributes (e.g., anatomical region, tissue type) that are predictable from the context. Conditioning on these variables simplifies density estimation, as conditional distributions are often less complex than marginal distributions.

### 3.3. Density model

The conditional density model $q_{\theta^{\mathrm{dens}}}(y \mid c)$ can be viewed as a predictive model, which tries to predict descriptors based on the corresponding conditions. In this interpretation, anomaly scores $\{-\log q_{\theta^{\mathrm{dens}}}(\mathbf{y}[p] \mid \mathbf{c}[p])\}_{p \in P}$ are position-wise prediction errors. Also note, that marginal density model $q_{\theta^{\mathrm{dens}}}(y)$ is a special case of conditional model with constant condition $\mathbf{c}[p] = \mathrm{const}$.

To train a conditional density model $q_{\theta^{\mathrm{dens}}}(y \mid c)$, we sample a batch of $m$ random crops, $\{\mathbf{x}_i\}_{i=1}^{m}$, each of size $H \times W \times$

*Table 1.* Summary information on the datasets that we use for training and testing of all models.

| Dataset | # 3D images | Annotated pathology |
|---|---|---|
| NLST (Team, 2011) | 25,652 | – |
| AMOS (Ji et al., 2022) | 2,123 | – |
| AbdomenAtlas (Qu et al., 2024) | 4,607 | – |
| LIDC (Armato III et al., 2011) | 1017 | lung cancer |
| MIDRC (Tsai et al., 2020) | 115 | pneumonia |
| KiTS (Heller et al., 2019) | 298 | kidney tumors |
| LiTS (Bilic et al., 2023) | 117 | liver tumors |

$S$, from different CT images. Each crop is passed through the pre-trained descriptor and condition models to produce descriptor maps, $\{\mathbf{y}_i\}_{i=1}^{m}$, and condition maps, $\{\mathbf{c}_i\}_{i=1}^{m}$. Then we optimize the conditional negative log-likelihood loss:

$$\min_{\theta_{\mathrm{dens}}} \quad \frac{1}{m \cdot |P|} \sum_{i=1}^{m} \sum_{p \in P} -\log q_{\theta^{\mathrm{dens}}}(\mathbf{y}_i[p] \mid \mathbf{c}_i[p]).$$

At inference, an input CT image is divided into $M$ overlapping patches, $\{\mathbf{x}_i\}_{i=1}^{M}$, each of size $H \times W \times S$. For each patch, we apply the descriptor, condition, and conditional density models to compute the anomaly map, $\{-\log q_{\theta^{\mathrm{dens}}}(\mathbf{y}_i[p] \mid \mathbf{c}_i[p])\}_{p \in P}$. These patch-wise anomaly maps are upsampled to $H \times W \times S$ and aggregated into a single anomaly map for the entire CT image by averaging predictions in patches' overlapping regions.

We explore two parameterizations for the density model: Gaussian, as a straightforward baseline, and normalizing flows, similar to (Gudovskiy et al., 2022; Zhou et al., 2024), as an expressive generative model enabling tractable density estimation. These parameterizations and the details of their implementation in the context of UVAS framework are further described in Appendix C.

## 4. Experiments & results

### 4.1. Datasets

We train all models on three CT datasets: NLST (Team, 2011), AMOS (Ji et al., 2022) and AbdomenAtlas (Qu et al., 2024). Note that we do not use any image annotations during training. Some of the datasets employed additional criteria for patients to be included in the study, i.e. age, smoking history, etc. Note that such large scale training datasets include diverse set of patients, implying presence of various pathologies.

We test all models on four datasets: LIDC (Armato III et al., 2011), MIDRC-RICORD-1a (Tsai et al., 2020), KiTS (Heller et al., 2019) and LiTS (Bilic et al., 2023).

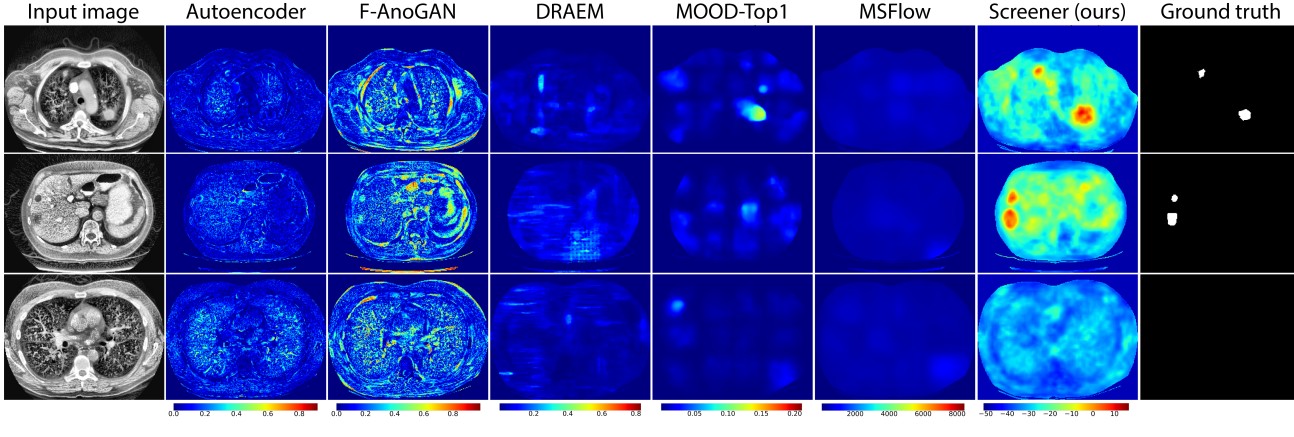

*Figure 3.* Qualitative comparison of anomaly maps produced by baseline UVAS methods and Screener. First column contains CT slices, columns 2 to 6 are baseline methods' predictions, column 7 is Screener's prediction. Last column depicts ground trught annotation mask.

*Table 2.* Quantitative comparison of Screener and the existing unsupervised visual anomaly segmentation methods on four test datasets with different pathologies.

| Model | AUROC | | | | AUROC up to FPR0.3 | | | | AUPRO up to FPR0.3 | | | |
|---|---|---|---|---|---|---|---|---|---|---|---|---|
| | LIDC | MIDRC | KiTS | LiTS | LIDC | MIDRC | KiTS | LiTS | LIDC | MIDRC | KiTS | LiTS |
| Autoencoder | 0.71 | 0.65 | 0.66 | 0.68 | 0.31 | 0.21 | 0.24 | 0.25 | 0.59 | 0.24 | 0.26 | 0.37 |
| f-AnoGAN | 0.82 | 0.66 | 0.67 | 0.67 | 0.52 | 0.21 | 0.24 | 0.22 | 0.46 | 0.18 | 0.24 | 0.22 |
| DRAEM | 0.63 | 0.72 | 0.82 | 0.83 | 0.21 | 0.31 | 0.50 | 0.51 | 0.17 | 0.20 | 0.50 | 0.57 |
| MOOD-Top1 | 0.79 | 0.79 | 0.77 | 0.80 | 0.43 | 0.43 | 0.40 | 0.46 | 0.32 | 0.29 | 0.40 | 0.32 |
| MSFlow | 0.70 | 0.66 | 0.64 | 0.64 | 0.26 | 0.20 | 0.18 | 0.17 | 0.21 | 0.14 | 0.19 | 0.17 |
| Screener (ours) | **0.96** | **0.87** | **0.90** | **0.93** | **0.88** | **0.64** | **0.68** | **0.80** | **0.65** | **0.40** | **0.67** | **0.63** |

Annotations of these datasets include segmentation masks of certain pathologies. Any other pathologies that can be present in these datasets are not labeled. We summarize the information about the datasets in Table 1.

### 4.2. Evaluation metrics

We use standard quality metrics for assessment of visual anomaly segmentation models which are employed in MVTecAD benchmark (Bergmann et al., 2021): pixel-level AUROC and AUPRO calculated up to 0.3 FPR. We also compute area under the whole pixel-level ROC-curve. Despite, our model can be viewed as semantic segmentation model, we do not report standard segmentation metrics, e.g. Dice score, due to the following reasons. As we mention in Section 4.1, available testing CT datasets contain annotations of only specific types of tumors, while other pathologies may be present in the images but not included in the ground truth masks. It makes impossible to fairly estimate metrics like Dice score or Hausdorff distance, which count our model's true positive predictions of the unannotated pathologies (see second image from the left in the Figure 1 for example) as false positive errors and strictly

penalize for them. However, the used pixel-level metrics are not sensitive to this issue, since they are based on sensitivity and specificity. We estimate sensitivity on pixels belonging to the annotated pathologies. To estimate specificity we use random pixels that do not belong to the annotated tumors which are mostly normal, thus yielding a practical estimate.

### 4.3. Main results

We compare Screener with baselines that represent different approaches to unsupervised visual anomaly segmentation. Specifically, we implement 3D versions of autoencoder (Baur et al., 2021), f-anoGAN (Schlegl et al., 2019) (reconstruction-based methods), DRAEM (Zavrtanik et al., 2021), MOOD-Top1 (Marimont & Tarroni, 2023) (methods based on synthetic anomalies) and MSFlow (density-based method on top of ImageNet features). Quantitative comparison is presented in Table 2. Qualitative comparison is shown in Figure 3.

The analysis of the poor performance of the reconstruction-based methods is given in Appendix E. Synthetic-based models yield many false negatives because during training they were penalized to predict zero scores in the unlabeled

*Table 3.* Ablation study of the effect of conditional model for the fixed descriptor model (VICReg) and different conditional density models (gaussian and normalizing flow). None in Condtion model column means that results are given for a marginal density model.

| Descriptor model | Condition model | Density model | AUROC | | | | AUROC up to FPR0.3 | | | | AUPRO up to FPR0.3 | | | |
|---|---|---|---|---|---|---|---|---|---|---|---|---|---|---|
| | | | LIDC | MIDRC | KiTS | LiTS | LIDC | MIDRC | KiTS | LiTS | LIDC | MIDRC | KiTS | LiTS |
| VICReg, $d^{desc} = 32$ | None | Gaussian | 0.81 | 0.81 | 0.61 | 0.71 | 0.41 | 0.47 | 0.12 | 0.22 | 0.46 | 0.62 | 0.13 | 0.28 |
| | Sin-cos pos. | Gaussian | 0.82 | 0.80 | 0.74 | 0.77 | 0.45 | 0.42 | 0.26 | 0.34 | 0.40 | 0.50 | 0.27 | 0.32 |
| VICReg, $d^{desc} = 32$ | APE | Gaussian | 0.88 | 0.80 | 0.78 | 0.86 | 0.67 | 0.46 | 0.34 | 0.56 | 0.43 | 0.38 | 0.35 | 0.55 |
| VICReg, $d^{desc} = 32$ | Masking-equiv. | Gaussian | **0.96** | **0.84** | **0.87** | **0.90** | **0.90** | **0.58** | **0.58** | **0.71** | **0.64** | **0.41** | **0.57** | **0.48** |
| VICReg, $d^{desc} = 32$ | None | Norm. flow | **0.96** | **0.89** | 0.88 | 0.93 | **0.89** | **0.68** | 0.62 | 0.78 | **0.67** | **0.46** | 0.62 | 0.65 |
| VICReg, $d^{desc} = 32$ | Sin-cos pos. | Norm. flow | **0.96** | **0.89** | 0.90 | 0.94 | **0.89** | **0.68** | 0.69 | 0.80 | 0.66 | **0.46** | 0.68 | 0.66 |
| VICReg, $d^{desc} = 32$ | APE | Norm. flow | **0.96** | 0.88 | 0.89 | **0.94** | 0.87 | 0.65 | 0.67 | **0.80** | 0.64 | 0.43 | 0.66 | **0.66** |
| VICReg, $d^{desc} = 32$ | Masking-equiv. | Norm. flow | **0.96** | 0.87 | **0.90** | 0.93 | 0.88 | 0.64 | 0.68 | **0.80** | 0.65 | 0.40 | 0.67 | 0.63 |

*Table 4.* Ablation study of the effect of descriptor model. In these experiments we do not use conditioning and use normalizing flow as a marginal density model. We include MSFlow to demonstrate that descriptor model pre-trained on ImageNet is inappropriate for 3D medical CT images.

| Descriptor model | Condition model | Density model | AUROC | | | | AUROC up to FPR0.3 | | | | AUPRO up to FPR0.3 | | | |
|---|---|---|---|---|---|---|---|---|---|---|---|---|---|---|
| | | | LIDC | MIDRC | KiTS | LiTS | LIDC | MIDRC | KiTS | LiTS | LIDC | MIDRC | KiTS | LiTS |
| ImageNet | Sin-cos pos. | MSFlow | 0.70 | 0.66 | 0.64 | 0.64 | 0.26 | 0.20 | 0.18 | 0.17 | 0.21 | 0.14 | 0.19 | 0.17 |
| STU-Net (Huang et al., 2023) | None | Norm. flow | 0.52 | 0.44 | 0.52 | 0.64 | 0.02 | 0.01 | 0.03 | 0.05 | 0.02 | 0.01 | 0.04 | 0.03 |
| SimCLR, $d^{desc} = 32$ | None | Norm. flow | **0.96** | 0.87 | **0.87** | 0.91 | **0.90** | 0.65 | 0.58 | 0.71 | 0.68 | 0.43 | 0.58 | 0.60 |
| VICReg, $d^{desc} = 32$ | None | Norm. flow | **0.96** | 0.89 | 0.88 | 0.93 | 0.89 | 0.68 | **0.62** | 0.78 | 0.67 | 0.46 | **0.62** | 0.65 |
| VICReg, $d^{desc} = 128$ | None | Norm. flow | **0.96** | **0.90** | 0.87 | 0.93 | **0.90** | **0.72** | 0.60 | **0.77** | **0.70** | **0.52** | 0.60 | **0.65** |

real pathological regions which may appear in training images. Meanwhile, MSFlow heavily relies on an ImageNet-pre-trained encoder which produces irrelevant features of 3D medical CT images. Our density-based model with domain-specific self-supervised features outperforms baselines by a large margin.

### 4.4. Condition and density models' ablation

Table 3 demonstrates ablation study of our proposed condition model. We compare our condition model with two baselines: vanilla sin-cos positional encodings and anatomical positional embeddings (Goncharov et al., 2024), described in Appendix B. We evaluate condition models in combination with different density models, described in Section 3.3. We use the VICReg descriptor model with $d^{desc} = 32$ as it shows slightly better results than contrastive objective as reported in Section 4.5.

When we use expressive normalizing flow density model, all conditioning strategies yield results comparable to each other and to the unconditional model. However, in experiments with simple Gaussian density models, we see that the results significantly improve as the conditioning variables becomes more informative. Noticeably, our proposed masking-invariant condition model allows Gaussian model to compete with complex flow-based models and achieve very strong anomaly segmentation results.

### 4.5. Descriptor models' ablation

We also ablate descriptor models in Table 4. We compare contrastive and VICReg models with $d^{desc} = 32$. To ablate the effect of the descriptors' dimensionality, we also include VICReg model with $d^{desc} = 128$. To demonstrate the superiority of our domain-specific self-supervised descriptors over supervised feature extractors pre-trained on natural images, we compare with MSFlow (Zhou et al., 2024). Additionally, we evaluate STU-Net (Huang et al., 2023) – a UNet pre-trained in a supervised manner on anatomical structure segmentation tasks – as a descriptor model in our framework. However, it performs even worse than MSFlow, likely because the feature maps from the penultimate UNet layer are too specific to the pre-training task and lack information about the presence of pathologies.

## 5. Related work

### 5.1. Visual unsupervised anomaly localization

In this section, we review several key approaches, each represented among the top five methods on the localization track of the MVTec AD benchmark (Bergmann et al., 2021), developed to stir progress in visual unsupervised anomaly detection and localization.

**Synthetic anomalies.** In unsupervised settings, real anomalies are typically absent or unlabeled in training images. To simulate anomalies, researchers synthetically cor-

rupt random regions by replacing them with noise, random patterns from a special set (Yang et al., 2023), or parts of other training images (Marimont & Tarroni, 2023). A segmentation model is trained to predict binary masks of corrupted regions, providing well-calibrated anomaly scores for individual pixels. While straightforward to train, these models may overfit to synthetic anomalies and struggle with real ones.

**Reconstruction-based.** Trained solely on normal images, reconstruction-based approaches (Baur et al., 2021; Kingma & Welling, 2013; Schlegl et al., 2019), poorly reconstruct anomalous regions, allowing pixel-wise or feature-wise discrepancies to serve as anomaly scores. Later generative approaches (Zavrtanik et al., 2021; Zhang et al., 2023; Wang et al., 2024) integrate synthetic anomalies. The limitation stemming from anomaly-free train set assumption still persists – if anomalous images are present, the model may learn to reconstruct anomalies as well as normal regions, undermining the ability to detect anomalies through differences between $x$ and $\hat{x}$.

**Density-based.** Density-based methods for anomaly detection model the distribution of the training image patterns. As modeling of the joint distribution of raw pixel values is infeasible, these methods usually model the marginal or conditional distribution of pixel-wise deep feature vectors.

Some methods (Roth et al., 2022; Bae et al., 2023) perform a non-parametric density estimation using memory banks. More scalable flow-based methods (Yu et al., 2021; Gudovskiy et al., 2022; Zhou et al., 2024), leverage normalizing flows to assign low likelihoods to anomalies. From this family, we selected MSFlow as a representative baseline, because it is simpler than PNI, and yields similar top-5 results on the MVTec AD.

### 5.2. Medical unsupervised anomaly localization

While there's no standard benchmark for pathology localization on CT images, MOOD (Zimmerer et al., 2022) offers a relevant benchmark with synthetic target anomalies. Unfortunately, at the time of preparing this work, the benchmark is closed for submissions, preventing us from evaluating our method on it. We include the top-performing method from MOOD (Marimont & Tarroni, 2023) in our comparison, that relies on synthetic anomalies.

Other recognized methods for anomaly localization in medical images are reconstruction-based: variants of AE / VAE (Baur et al., 2021; Shvetsova et al., 2021), f-AnoGAN (Schlegl et al., 2019), and diffusion-based (Pinaya et al., 2022). These approaches highly rely on the fact that the the training set consists of normal images only. However, it is challenging and costly to collect a large dataset of

CT images of normal patients. While these methods work acceptable in the domain of 2D medical images and MRI, the capabilities of the methods have not been fully explored in a more complex CT data domain. We have adapted these methods to 3D.

## 6. Conclusion

This work explores a fully self-supervised approach to pathology segmentation in 3D medical images using a density-based UVAS framework. Existing UVAS methods rely on anomaly-free training datasets or supervised feature extractors, which are unavailable for CT images. To address these limitations, we introduce Screener, extending the density-based UVAS framework with two key innovations: (1) a self-supervised representation learning descriptor for image features, and (2) a trainable conditioning model that enhances simpler density models. Screener, being domain-specific and self-supervised, overcomes the limitations of earlier methods and achieves superior performance, as demonstrated by our empirical results.

**Limitations.** This work serves as a proof-of-concept for two hypotheses: (1) pathology segmentation in CT images can be approached as UVAS, and (2) density estimation in dense self-supervised feature spaces yields meaningful anomaly scores. However, unsupervised approach inevitably has limitations. Statistically abnormal visual patterns do not always align with clinically significant abnormalities, leading to unavoidable false positives and negatives. Additionally, our training dataset is biased toward chest CTs, resulting in more false positives in abdominal regions. Generalization to other anatomical regions requires training on corresponding datasets.

**Future work.** While the performance gains compared to baselines are already significant, we note that further improvements might be achieved from increasing descriptors and conditions dimensionality and experiments with multi-scale representations (e.g. by building feature pyramids). Another possible avenue for future work is to study scaling laws, i.e. self-supervised models typically scale well with increasing pre-training dataset sizes. Distillation of Screener into UNet and subsequent supervised fine-tuning is also an interesting practical application of our work but needs further exploration.

## Impact Statement

This paper presents work whose goal is to advance the field of Machine Learning. There are many potential societal consequences of our work, none which we feel must be specifically highlighted here.

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

## A. Self-Supervised Learning

**InfoNCE.** In contrastive learning, batch of positive pairs $\{(y_i^{(1)}, y_i^{(2)})\}_{i=1}^N$ is passed through a trainable MLP-projector $g_{\theta^{\text{proj}}}$ and l2-normalized: $z_i^{(k)} = g_{\theta^{\text{proj}}}(y_i^{(k)})/\|g_{\theta^{\text{proj}}}(y_i^{(k)})\| \in \mathbb{R}^d$, where $k = 1, 2$ and $i = 1, \ldots N$. Then, the key objective is to maximize the similarity between embeddings of positive pairs while minimizing their similarity with negative pairs. To this end, InfoNCE loss written as:

$$\min_{\theta} \quad \sum_{i=1}^N \sum_{k \in \{1,2\}} - \log \frac{\exp(\langle z_i^{(1)}, z_i^{(2)} \rangle / \tau)}{\exp(\langle z_i^{(1)}, z_i^{(2)} \rangle / \tau) + \sum_{j \neq i} \sum_{l \in \{1,2\}} \exp(\langle z_i^{(k)}, z_j^{(l)} \rangle / \tau)}. \tag{1}$$

**VICReg.** VICReg objective enforces invariance among positive embeddings while constraining embeddings' covariance matrix to be diagonal and variance to be equal to some constant:

$$\min_{\theta} \quad \alpha \cdot \mathcal{L}^{\text{inv}} + \beta \cdot \mathcal{L}^{\text{var}} + \gamma \cdot \mathcal{L}^{\text{cov}}. \tag{2}$$

The first term $\mathcal{L}^{\text{inv}} = \frac{1}{N \cdot D} \sum_{i=1}^N \|z_i^{(1)} - z_i^{(2)}\|^2$ penalizes embeddings to be invariant to augmentations. The second term $\mathcal{L}^{\text{var}} = \sum_{k \in \{1,2\}} \frac{1}{D} \sum_{i=1}^D \max\left(0, 1 - \sqrt{C_{i,i}^{(k)} + \varepsilon}\right)$ enforces individual embeddings' dimensions to have unit variance. The third term $\mathcal{L}^{\text{cov}} = \sum_{k \in \{1,2\}} \frac{1}{D} \sum_{i \neq j} \left(C_{i,j}^{(k)}\right)^2$ encourages different embedding's dimensions to be uncorrelated, increasing the total information content of the embeddings. In VICReg embeddings $\{z_i^{(k)}\}$ are not l2-normalized and obtained through a trainable MLP-expander which increases the dimensionality up to $8192$.

## B. Baseline condition models

**Sin-cos positional encodings.** The existing density-based UVAS methods (Gudovskiy et al., 2022; Zhou et al., 2024) for natural images use standard sin-cos positional encodings for conditioning. We also employ them as an option for condition model in our framework. However, let us clarify what we mean by sin-cos positional embeddings in CT images. Note that we never apply descriptor, condition or density models to the whole CT images due to memory constraints. Instead, at all the training stages and at the inference stage of our framework we always apply them to image crops of size $H \times W \times S$, as described in Sections 3.1, 3.3. When we say that we apply sin-cos positional embeddings condition model to an image crop, we mean that compute sin-cos encodings of absolute positions of its pixels w.r.t. to the whole CT image.

**Anatomical positional embeddings.** To implement the idea of learning the conditional distribution of image patterns at each certain anatomical region, we need a condition model producing conditions $c[p]$ that encode which anatomical region is present in the image at every position $p$. Supervised model for organs' semantic segmentation would be an ideal condition model for this purpose. However, to our best knowledge, there is no supervised models that are able to segment all organs in CT images. That is why, we decided to try the self-supervised APE (Goncharov et al., 2024) model which produces continuous embeddings of anatomical position of CT image pixels.

## C. Density Models

Below, we describe simple Gaussian density model and more expressive learnable Normalizing Flow model.

**Gaussian** marginal density model is written as

$$- \log q_{\theta^{\text{dens}}}(y) = \frac{1}{2}(y - \mu)^\top \Sigma^{-1}(y - \mu) + \frac{1}{2} \log \det \Sigma + \text{const}, \tag{3}$$

where the trainable parameters $\theta^{\text{dens}}$ are mean vector $\mu$ and diagonal covariance matrix $\Sigma$.

Conditional gaussian density model is written as

$$- \log q_{\theta^{\text{dens}}}(y \mid c) = \frac{1}{2}(y - \mu_{\theta^{\text{dens}}}(c))^\top (\Sigma_{\theta^{\text{dens}}}(c))^{-1} (y - \mu_{\theta^{\text{dens}}}(c)) + \frac{1}{2} \log \det \Sigma_{\theta^{\text{dens}}}(c) + \text{const}, \tag{4}$$

where $\mu_{\theta^{\text{dens}}}$ and $\Sigma_{\theta^{\text{dens}}}$ are MLP nets which take condition $c \in \mathbb{R}^{d^{\text{cond}}}$ as input and predict a conditional mean vector $\mu_{\theta^{\text{dens}}}(c) \in \mathbb{R}^{d^{\text{desc}}}$ and a vector of conditional variances which is used to construct the diagonal covariance matrix $\Sigma_{\theta^{\text{dens}}}(c) \in \mathbb{R}^{d^{\text{desc}} \times d^{\text{desc}}}$.

As described in Section 3.3, at both training and inference stages, we need to obtain dense negative log-density maps. Dense prediction by MLP nets $\mu_{\theta^{\text{dens}}}(c)$ and $\Sigma_{\theta^{\text{dens}}}(c)$ can be implemented using convolutional layers with kernel size $1 \times 1 \times 1$. In practice, we increase this kernel size to $3 \times 3 \times 3$, which can be equivalently formulated as conditioning on locally aggregated conditions.

**Normalizing flow** model of descriptors' marginal distribution is written as:

$$-\log p_{\theta^{\text{dens}}}(y) = \frac{1}{2}\|f_{\theta^{\text{dens}}}(y)\|^2 - \log\left|\det \frac{\partial f_{\theta^{\text{dens}}}(y)}{\partial y}\right| + \text{const}, \tag{5}$$

where neural net $f_\theta$ must be invertible and has a tractable jacobian determinant.

Conditional normalizing flow model of descriptors' conditional distribution is given by:

$$-\log p_{\theta^{\text{dens}}}(y \mid c) = \frac{1}{2}\|f_{\theta^{\text{dens}}}(y, c)\|^2 - \log\left|\det \frac{\partial f_{\theta^{\text{dens}}}(y, c)}{\partial y}\right| + \text{const}, \tag{6}$$

where neural net $f_\theta \colon \mathbb{R}^{d^{\text{desc}}} \times \mathbb{R}^{d^{\text{cond}}} \to \mathbb{R}^{d^{\text{desc}}}$ must be invertible w.r.t. the first argument, and the second term should be tractable.

We construct $f_\theta$ by stacking Glow layers (Kingma & Dhariwal, 2018): act-norms, invertible linear transforms and affine coupling layers. Note that at both training and inference stages we apply $f_\theta$ to descriptor maps $\mathbf{y} \in \mathbb{R}^{h \times w \times s \times d^{\text{desc}}}$ in a pixel-wise manner to obtain dense negative log-density maps. In conditional model, we apply conditioning in affine coupling layers similar to (Gudovskiy et al., 2022) and also in each act-norm layer by predicting maps of rescaling parameters based on condition maps.

# D. Other implementation details

For our Screener model, we pre-process CT volumes by cropping them to dense foreground voxels (thresholded by $-500$HU), resizing to $1.5 \times 1.5 \times 2.25$ mm$^3$ voxel spacing, clipping intensities to $[-1000, 300]$HU and rescaling them to $[0, 1]$ range. As an important final step we apply CLAHE (Pizer et al., 1987). CLAHE ensures that color jitter augmentations preserve information about presence of pathologies during descriptor model training (otherwise, the quality of our method degrades largely).

We train both the descriptor model and the condition model for 300k batches of $m = 8$ pairs of overlapping patches with $N = 8192$ positive pairs of voxels. The training takes about 3 days on a single NVIDIA RTX H100-80GB GPU. We use AdamW optimizer, warm-up learning rate from 0.0 to 0.0003 during first 10K batches, and then reduce it to zero till the end of the training. Weight decay is set to $10^{-6}$ and gradient clipping to 1.0 norm. Patch size is set to $H \times W \times S = 96 \times 96 \times 64$.

During the density model training we apply average pooling operations with $3 \times 3 \times 2$ stride to feature maps produced by the descriptor model as well as the condition model, following (Gudovskiy et al., 2022; Zhou et al., 2024). Thus $h \times w \times s = 32 \times 32 \times 32$. We inject gaussian noise with 0.1 standard deviation both to the descriptors and to the conditions in order to stabilize the training. We train the density model for 500k batches each containing $m = 4$ patches. This training stage again takes about 3 days on a single NVIDIA RTX H100-80GB GPU. We use the same optimizer and the learning rate scheduler as for the descriptor and condition models.

# E. Analysis of reconstruction-based models

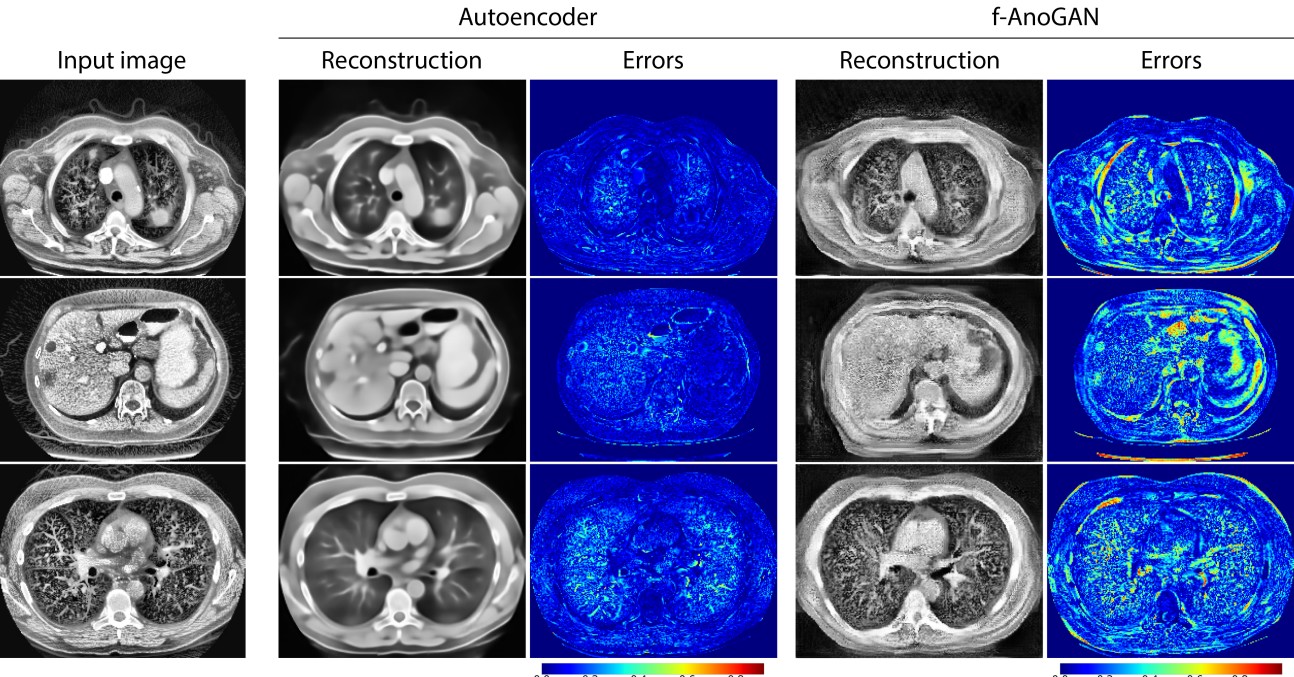

Figure 4. The figure shows 2D slices of CT images (first column) alongside reconstructions and anomaly maps generated by two methods: an Autoencoder (Baur et al., 2021) (second and third columns) and f-AnoGAN (Schlegl et al., 2019) (last two columns). Autoencoder overfits for pixel reconstruction, so it generates pathologies and fails to segment them. Also Autoencoder produces blurry generations, leading to inaccurate reconstructions of fine details and high anomaly scores on these details (e.g., vessels in the lungs). f-AnoGAN, on the other hand, avoids generating pathologies, but the generation quality still is insufficient for precise segmentation of only pathological voxels. GANs are known to be unstable and sensitive to hyperparameters, necessitating careful tuning and experimentation to achieve optimal results.

