# OpenReview forum: "Screener: Self-supervised Pathology Segmentation Model for 3D Medical Images"
_ICML.cc/2025/Conference — Submitted to ICML 2025_

### Official Review · Reviewer_pFWf · 2025-03-11

**Overall Recommendation:** 3

**Summary:**

In this paper, an unsupervised visual anomaly detection algorithm is proposed, which the authors describe as a segmentation algorithm, although I disagree with this characterization. The method exploits the inherent rarity of pathological patterns compared to healthy ones. Two different self-supervised learning strategies are employed to train a descriptor and a condition model. The outputs of these models are then used to train a density model that generates voxel-wise anomaly scores. The model, trained on over 30,000 unlabeled 3D CT volumes, appears to outperform existing methods on four test datasets, comprising 1,820 scans with diverse pathologies.

**Claims And Evidence:**

The authors claim to introduce a pathology segmentation algorithm designed for accurate segmentation of all pathological findings in 3D medical images, with the ability to handle pathology classes beyond those in the training datasets. They also claim to reframe pathology segmentation as an unsupervised visual anomaly segmentation problem. However, I believe it is inaccurate to describe the algorithm as a segmentation model, and there is no actual reframing—the algorithm is designed for anomaly detection. Additionally, while I noticed that disjoint data are used for training and testing, this doesn't mean novel pathology classes exist in the testing data and the first claim is well justified. Do testing images include novel pathology classes not present in the training data?

**Essential References Not Discussed:**

Some recent related works, such as [1], are not mentioned or compared. It remains unclear whether the proposed method outperforms them and to what extent improvement is achieved.

[1] Huang, Chaoqin, et al. "Adapting visual-language models for generalizable anomaly detection in medical images." Proceedings of the IEEE/CVF Conference on Computer Vision and Pattern Recognition. 2024.

**Experimental Designs Or Analyses:**

My concerns are: 1) Whether the claim that the proposed method can be used for novel anomaly detection is justified, and 2) that the algorithm is for anomaly detection rather than segmentation. For an anomaly detection algorithm evaluation, the experimental design and evaluation metric seem okay.

**Methods And Evaluation Criteria:**

Yes

**Other Comments Or Suggestions:**

1. I believe the proposed method can only be considered a segmentation method if its final output is a segmentation mask. I encourage reconsidering the use of terms like ‘segment’ and ‘segmentation’ throughout the paper.

2. I suggest analyzing the differences between the outputs of the descriptor and condition models and providing more insights and visualizations to help readers better understand the rationale.

**Other Strengths And Weaknesses:**

The idea is interesting and the results seem promising. However, I have the following two concerns that need to be addressed:

1. It is unclear what assumptions are made about the training data but not explicitly stated. Does the training data need to be dominated by images without pathology?

2. It is unclear how different the features extracted by the descriptor and condition models are and if different, whether one can be inferred from the other, which is crucial for training the density model. My concern is that if the input feature pairs are significantly different, how can one be reliably inferred from the other? Conversely, if they are too similar, the output of density model would be less meaningful.

**Questions For Authors:**

1. Are there any assumptions about the training data? Would the method still work with a dataset consisting entirely of images with pathology?

2. It is mentioned that "the descriptor model must generate descriptors that effectively differentiate between pathological and normal positions." However, how this is guaranteed with the adopted training strategy?

3. How different are the features extracted by the descriptor and condition models? How reliably can the feature from the condition model be used to infer the feature from the descriptor model? Have you compared the inferred features with the extracted ones? My concern is that if the input feature pairs are significantly different, how can one be reliably inferred from the other? Conversely, if they are too similar, the density model’s output may be less meaningful, calling into question the rationale of the proposed method.

4. How did you determine the patch size for training? Would a smaller or larger patch size work as well? Specifically, I am interested in how the patch size affects the differences between the features extracted by the descriptor and condition models.

**Relation To Broader Scientific Literature:**

This paper is closely related to anomaly detection algorithms like DRAEM in both computer vision and medical imaging domain.

**Theoretical Claims:**

N/A

---

> ### Author Rebuttal · Authors · 2025-04-01
>
> Dear Reviewer *pFWf*, thank you for your thorough review and valuable feedback. We appreciate your thoughtful questions and have carefully addressed each point below.
>
> **Terminology: anomaly detection vs. segmentation**
>
> We agree that terminology in this field can vary. In our work we use **"anomaly segmentation"** to refer specifically to pixel-level anomaly detection, **consistent with established literature [1-3]**. While supervised segmentation models produce probability maps, our method generates **anomaly score maps** that **can be thresholded to obtain segmentation masks**.
>
> To better align with medical image segmentation evaluation standards, **we have added Dice scores to Table 2 (https://pdfhost.io/v/n7Bt5c7zJE_table_2) and Tables 3-4 (https://pdfhost.io/v/2RLGCYvMgy_tables_3_4)**.
>
> Initially, we omitted Dice scores due to a mismatch between our problem statement (segmenting all pathologies) and the available ground truth masks (limited to specific pathologies — lung cancer in LIDC, pneumonia in MIDRC, liver tumors in LiTS, kidney tumors in KiTS). Note that this discrepancy leads to an **underestimation** of unsupervised models’ Dice scores, as many test images indeed contain additional anomalies detected by our model but not annotated in the ground truth masks (see Figure 2 and more examples at https://pdfhost.io/v/PNygYvbsYn_mismatch which we will add in the Appendix).
>
> **Training data assumptions**
>
> **Our key assumption:** *each pathological pattern is rarer (has lower density in an embedding space) than any normal pattern*. This holds true when abnormal patterns are diverse (few similar cases), while normal patterns recur frequently across patients.
>
> In practice, our training data distribution presents a mixture of 25K chest CTs of screening patients (NLST) and 7K abdominal CTs from hospitals (AMOS, AbdomenAtlas). Our empirical results show that **our model is capable of detecting both chest and abdominal pathologies, despite being trained on such an imbalanced and uncurated dataset**.
>
> **Is Screener capable of detecting novel pathologies?**
>
> **Theoretically, Screener is capable of detecting novel pathologies** which are not present in training dataset, because in this case, there density is almost zero (not exactly zero due to the addition of gaussian noise during training), and Screener will assign large negative log density scores to them.
>
> Our **current evaluation uses pathologies (lung cancer, pneumonia, liver/kidney tumors) that likely exist in our training data**. While this doesn't demonstrate novel-class detection, it shows generalization across diverse manifestations of these pathologies.
>
> A compelling test would involve training on pre-pandemic data and evaluating on COVID-19 lesions. We will propose this as important future work in Section 6.
>
> **Are descriptor model features inferable from condition model features?**
>
> Descriptor model and condition model are trained separately, and produce different pixel-level feature vectors, denoted as $y$ (descriptors) and $c$ (conditions).
>
> Theoretically, there is no functional relation between descriptors and conditions: two different pixels may have different descriptors and the same condition. However, our density model learns the conditional density $q(y \mid c)$ of descriptors for every given condition. Intuitively, if $-\log q(y \mid c)$ is large, the observed $y$ has low probability according to the conditional distribution, and is treated as anomaly. This intuition is well supported by our both quantitative and qualitative empirical results.
>
> **How do we ensure the descriptor model to differentiate between pathological and normal regions?**
>
> We pre-train our descriptor model using dense VICReg loss (Section 3, Appendix A). VICReg’s regularization encourages the descriptors' covariance matrix to be near identity, ensuring **feature maps are non-trivial** (unit variance along spatial dimensions) and **uncorrelated** (distinct channels capture different features). This helps descriptors distinguish between different pixels, particularly pathological and normal ones.
>
> **Patch size selection**
>
> We selected patch size and voxel spacing based on our previous experience with supervised pathology segmentation models.
>
> We hope that we have answered your questions and addressed your concerns. Please let us know if further clarifications are needed.
>
> **References**
>
> [1] Bergmann, Paul, et al. "MVTec AD--A comprehensive real-world dataset for unsupervised anomaly detection." Proceedings of the IEEE/CVF conference on computer vision and pattern recognition. 2019.
>
> [2] Ghorbel, Ahmed, et al. "Transformer based models for unsupervised anomaly segmentation in brain MR images." International MICCAI Brainlesion Workshop. Cham: Springer Nature Switzerland, 2022.
>
> [3] Zou, Yang, et al. "Spot-the-difference self-supervised pre-training for anomaly detection and segmentation." European Conference on Computer Vision. Cham: Springer Nature Switzerland, 2022.

---

### Official Review · Reviewer_hWnx · 2025-03-12

**Overall Recommendation:** 3

**Summary:**

The authors introduce Screener, a self-supervised 3D pathology segmentation model that formulates the task as an unsupervised visual anomaly segmentation (UVAS) problem. It utilizes self-supervised feature learning and a masking-invariant condition model within a density-based UVAS framework. Trained on 30,000+ unlabeled CT scans, Screener is evaluated on 1,820 test scans across four datasets, achieving AUROC up to 0.96. The study conducts a large-scale evaluation of UVAS for 3D CT images and explores self-supervised learning for medical image pathology segmentation.

**Claims And Evidence:**

The paper provides quantitative and qualitative evidence to support its claims through experiments, ablation studies, and comparisons with multiple baseline methods.

**Essential References Not Discussed:**

The authors may refer more to self-supervised medical image segmentation, e.g.,[1] and [2].


[1] Tang, Yucheng, et al. "Self-supervised pre-training of swin transformers for 3d medical image analysis." Proceedings of the IEEE/CVF conference on computer vision and pattern recognition. 2022.

[2] Zhou, Hong-Yu, et al. "A unified visual information preservation framework for self-supervised pre-training in medical image analysis." IEEE Transactions on Pattern Analysis and Machine Intelligence 45.7 (2023): 8020-8035.

**Experimental Designs Or Analyses:**

It is better to provide some supervised baseline for reference—comparison with a fully supervised segmentation model could help contextualize how much performance is lost by using UVAS.

**Methods And Evaluation Criteria:**

The evaluation criteria are generally appropriate but warrant some scrutiny:

- AUROC and AUPRO are standard for anomaly detection tasks, making them fitting choices given the UVAS framing. They effectively evaluate the model’s ability to detect rare pathological pixels, which aligns with the problem’s focus on identifying deviations from normal tissue. However, traditional segmentation metrics like the Dice coefficient or Jaccard index, which measure overlap between predicted and ground truth segments, are more common in clinical segmentation tasks. These metrics provide direct interpretability for clinicians (e.g., “How much of the tumor was correctly segmented?”), which AUROC and AUPRO do not. Including both anomaly detection and segmentation metrics would offer a more holistic assessment.

- Outperforming existing UVAS methods demonstrates the effectiveness of Screener’s innovations. However, a comparison with supervised methods (where labeled data is available) would contextualize the performance gap, providing insight into how close the self-supervised approach comes to fully supervised benchmarks—a relevant consideration for clinical adoption.

**Other Comments Or Suggestions:**

NA

**Other Strengths And Weaknesses:**

Strengths:
- The paper extends density-based unsupervised visual anomaly segmentation (UVAS) to 3D CT pathology segmentation, an area with limited prior work.

- Trains on 30,000+ unlabeled CT scans and evaluates on 1,820 labeled scans from four pathology datasets, providing strong empirical validation.

Weaknesses:
- While the paper compares against unsupervised baselines, it does not include a fully supervised segmentation model (e.g., UNet trained on labeled data). This makes it difficult to quantify how much performance loss occurs due to using UVAS instead of a supervised approach.

- The model is trained on high-resolution 3D CT scans, but the paper does not discuss computational cost or inference speed, which are critical for clinical deployment.

- The authors may try to improve the writing of this paper, especially for the method sections (e.g., notations and clarity of descriptions of concepts).

**Questions For Authors:**

Please try to address the weaknesses.

**Relation To Broader Scientific Literature:**

The key contributions of the paper build on and extend several existing approaches in self-supervised learning, anomaly detection, and medical image segmentation.

- Traditional supervised segmentation models rely on large labeled datasets (e.g., UNet, Ronneberger et al., 2015), which are scarce for medical imaging.

- Self-supervised learning (SSL) has been successfully applied to natural images (e.g., SimCLR, VICReg), but its application to 3D CT medical images is less explored. This work uses dense self-supervised learning for 3D CT volumes.

**Theoretical Claims:**

NA

---

> ### Author Rebuttal · Authors · 2025-04-01
>
> Dear Reviewer *hWnx*, thank you for taking the time to review our submission and for providing thoughtful and valuable feedback. Your suggestions regarding our evaluation design were especially valuable, and we have done our best to accomplish them, as well as address your other concerns.
>
> **Inclusion of Dice scores**
>
> To provide more interpretable metrics **we have updated Table 2 (https://pdfhost.io/v/n7Bt5c7zJE_table_2) and Tables 3, 4 (https://pdfhost.io/v/2RLGCYvMgy_tables_3_4) to include Dice scores and voxel-level AUROCs**. In order to improve tables’ readability, we decided to move AUROC / AUPRO up to 0.3 FPR metrics to the Appendix.
>
> Initially, we omitted Dice scores due to a mismatch between our problem statement (segmenting all pathologies) and the available ground truth masks (limited to specific pathologies — lung cancer in LIDC, pneumonia in MIDRC, liver tumors in LiTS, kidney tumors in KiTS). Note that this discrepancy leads to an **underestimation** of unsupervised models’ Dice scores, as many test images indeed contain additional anomalies detected by our model but not annotated in the ground truth masks (see Figure 2 and more examples at https://pdfhost.io/v/PNygYvbsYn_mismatch which we will add in the Appendix).
>
> **Comparison with supervised baseline**
>
> **We have added comparison with a Supervised UNet (Table 2, https://pdfhost.io/v/n7Bt5c7zJE_table_2)**, trained via cross validation on each dataset (LIDC, MIDRC, LiTS and KiTS). Since the supervised model segments only the target pathology, its metrics are naturally higher than those of unsupervised models.
>
> For a fair comparison, **we distilled Screener into UNet and fine-tuned it in a supervised manner (Fine-tuned Screener)**. Namely, at the distillation step, we pre-train UNet (without final sigmoid activation) to predict Screener’s anomaly score maps (using simple MSE loss). Then, at the supervised fine-tuning (SFT) stage, we randomly re-initialize the pre-trained UNet’s last conv layer and train it on each dataset similarly to Supervised UNet. **Table 2 (https://pdfhost.io/v/n7Bt5c7zJE_table_2) shows that Fine-tuned Screener consistently outperforms Supervised UNet**, especially on lung cancer segmentation.
>
> To demonstrate the significance of the latter result we also have drawn plots (https://pdfhost.io/v/yeejsw47bF_train_sizes) comparing Supervised UNet and Fine-tuned screener trained on datasets’ subsamples of different sizes (10, 20 or 40 images per train fold). Note that **when training on 20 images with annotated lung cancer, Fine-tuned Screener achieves 2 times higher Dice scores than Supervised UNet**.
>
> **Inference speed & computational cost**
>
> Our original Screener model has 133M parameters, patch-based inference for a whole CT volume (described in Section 3.3) on NVIDIA RTX H100 GPU requires 4 Gb of GPU memory and takes about 5-10 seconds depending on the number of slices.
>
> Also, as discussed above, Screener can be distilled into the standard UNet model (we did not observe significant changes in quality metrics for the distilled model). Thus, its inference costs can be the same as those of UNet. We use nnUNet with 350M parameters, its patch-based inference requires **5 Gb of GPU memory** and takes **0.5-1.0 seconds**. We will include this information in the Implementation details section.
>
> **Referring to existing SSL models for CT**
>
> We will add the suggested references in Section 5.
>
> **Improving paper writing**
>
> We will improve paper writing, for example, use consistent methods naming in Sections 3 and 4, use consistent math notation, improve writing in Section 5, etc.
>
> We sincerely hope these revisions address most of your concerns. Please let us know if further clarifications are needed to reconsider the score.

---

### Official Review · Reviewer_X8PP · 2025-03-14

**Overall Recommendation:** 4

**Summary:**

This paper proposed Screener, a self-supervised anomaly segmentation framework for volumetric CT images.
The Screener was built upon dense self-supervised learning and a density-based anomaly segmentation framework. Specifically, it utilizes dense pixel-wise self-supervised learning (i.e., VICReg) to pretrain two encoders serving as descriptor and condition models. A density model (Gaussian or normalizing flow) takes the joint embedding to estimate and assign pixel-wise anomaly scores. The model was pretrained on large-scale 30k CT scans and was evaluated on four different CT datasets and outperformed other baseline methods.

**Claims And Evidence:**

The claims of contribution are mostly supported by convincing evidence. However, the value of conditioning variables is arguable as it does not affect the performance when using normalization flow as the density model.

**Essential References Not Discussed:**

Recent medical anomaly segmentation studies [1,3, 4] focus on the same problem (medical anomaly segmentation), and are more recent developments compared to f-AnoGAN discussed in the manuscript. [4] also utilizes normalizing flow, the same as the proposed method.

[2, 5] are for the natural images but represent the most recent development. [5] also explore utilizing self-supervised pretraining for anomaly segmentation.

[1]: Pinaya, Walter HL, et al. "Unsupervised brain imaging 3D anomaly detection and segmentation with transformers." Medical Image Analysis 79 (2022): 102475.

[2]: Liu, Zhikang, et al. "Simplenet: A simple network for image anomaly detection and localization." Proceedings of the IEEE/CVF conference on computer vision and pattern recognition. 2023.

[3]: Iqbal, Hasan, et al. "Unsupervised anomaly detection in medical images using masked diffusion model." International Workshop on Machine Learning in Medical Imaging. Cham: Springer Nature Switzerland, 2023.

[4]: Zhao, Yuzhong, Qiaoqiao Ding, and Xiaoqun Zhang. "AE-FLOW: Autoencoders with normalizing flows for medical images anomaly detection." The Eleventh International Conference on Learning Representations. 2023.

[5]: Zou, Yang, et al. "Spot-the-difference self-supervised pre-training for anomaly detection and segmentation." European Conference on Computer Vision. Cham: Springer Nature Switzerland, 2022.

**Experimental Designs Or Analyses:**

The reviewer found the experimental designs not comprehensive enough. Although the authors listed a few representative methods from different perspectives (i.e., synthetic anomalies, recon-based, density-based for nature image, and domain-specific medical unsupervised anomaly localization), the current manuscript misses a few of the most recent studies [1-5]. For example, the f-AnoGAN, a 2019 baseline, is the only one specifically designed for medical images in experiments. This incomplete baseline comparison weakens the convincingness of the results.

[1]: Pinaya, Walter HL, et al. "Unsupervised brain imaging 3D anomaly detection and segmentation with transformers." Medical Image Analysis 79 (2022): 102475.

[2]: Liu, Zhikang, et al. "Simplenet: A simple network for image anomaly detection and localization." Proceedings of the IEEE/CVF conference on computer vision and pattern recognition. 2023.

[3]: Iqbal, Hasan, et al. "Unsupervised anomaly detection in medical images using masked diffusion model." International Workshop on Machine Learning in Medical Imaging. Cham: Springer Nature Switzerland, 2023.

[4]: Zhao, Yuzhong, Qiaoqiao Ding, and Xiaoqun Zhang. "AE-FLOW: Autoencoders with normalizing flows for medical images anomaly detection." The Eleventh International Conference on Learning Representations. 2023.

[5]: Zou, Yang, et al. "Spot-the-difference self-supervised pre-training for anomaly detection and segmentation." European Conference on Computer Vision. Cham: Springer Nature Switzerland, 2022.

**Methods And Evaluation Criteria:**

The methods and evaluation criteria make sense.

**Other Comments Or Suggestions:**

1. The visualization of Figure 3 should be corrected. The color bar for other methods looks very strange; for example, MSFlow has a color bar from 0 to 8000.

**Other Strengths And Weaknesses:**

The overall results look superior to the listed baseline methods, although a few more recent and relevant studies are not included in the experiments.

The presentation could be improved and the logical flow can be optimized (e.g., move related work to an earlier position).

**Questions For Authors:**

Why not consider other self-supervised pretext tasks besides SimCLR and VICReg, such as masked autoencoding?

**Relation To Broader Scientific Literature:**

This paper explores using self-supervised learning to enhance the density-based anomaly segmentation framework, focusing on medical images. It is related to previous studies focusing on 3D medical anomaly segmentation, density-based anomaly segmentation, and self-supervised learning for anomaly segmentation.

**Theoretical Claims:**

There is no proof of any theoretical claims.

---

> ### Author Rebuttal · Authors · 2025-04-01
>
> Dear Reviewer *X8PP*, thank you for your careful review and constructive feedback. We appreciate your acknowledgment of our contributions and have carefully addressed your suggestions below.
>
> **Inclusion of recent baselines**
>
> We recognize the value of benchmarking our approach against recent state-of-the-art methods. Below, we address each of the provided references:
> - **Masked Diffusion Model [3]**: We are now re-implementing this method for CT images and **will include it in Table 2 during the discussion period**.
> - **Simplenet [2]**: We are also implementing Simplenet (inspired by noise-contrastive methods for unnormalized density estimation) on top of our descriptor model and **will add it to Table 3** for comparison with our flow-based and gaussian density models.
> - **Transformer-based method [1]**: While we recognize the importance of [1], its implementation (requiring VQ-VAE pretraining, transformer-based autoregressive modeling of VQ-VAE latent codes, and likelihood threshold tuning) is too complex to complete within the rebuttal period. We will, however, discuss it in Section 5.
>
> The other two methods are not directly applicable to our setup:
> - **AE-FLOW [4]**: We note that [4] focuses on *image-level* anomaly detection, whereas our work targets *pixel-level* anomaly segmentation.
> - **Spot-the-difference [5]**: Though this self-supervised strategy (penalizing features’ sensitivity to synthetic anomalies) is relevant, [5] evaluates it on *supervised* anomaly detection. We will discuss its potential applicability to our dense SSL framework and unsupervised anomaly segmentation in Section 6 as future work.
>
> **MSFlow colorbar in Figure 3**
>
> Thank you for catching this issue. Following your remark, we have fixed our MSFlow implementation and updated its presentation in Table 2 (https://pdfhost.io/v/n7Bt5c7zJE_table_2) and Figure 3 (https://pdfhost.io/v/NHDraEfxhJ_main_results).
>
> **Alternative SSL strategies for descriptor and condition models**
>
> As described in Section 3, we employed dense joint embedding SSL methods because they offer a unified framework for training both descriptor and condition models and allow us to control the information content of the learned features by changing the augmentations. Namely, augmentations preserving local content ensure that the descriptor model captures pathology-aware features, while random masking results in the pathology-ignorant condition model. We agree that exploring other SSL strategies (e.g., masked autoencoding) is promising and will mention this direction in Section 6.
>
> **Positioning of Related Work section**
>
> Our current structure aims to balance clarity and emphasis on novelty:
> - **Section 2 (Background):** Discusses related works directly inspiring our method.
> - **Section 5 (Related Work):** Provides a broader review of UVAS families.
>
> We believe this flow better highlights our methodological contributions.
>
> We hope that our revisions will address your main concerns. If there are any other areas where you feel further improvements can be made, we would be grateful for your additional feedback.

---

> > ### Comment · Reviewer_X8PP · 2025-04-03
> >
> > Thank you to the authors for providing the detailed rebuttal and additional experiments. After reading everyone's comments and the corresponding rebuttal, I think the paper has been improved. Therefore, I will raise the score to 4: Accept.

---

> > > ### Author Response · Authors · 2025-04-06
> > >
> > > Thank you for your patience and for the opportunity to expand our experiments. Following your suggestion, we have now included two additional recent baselines.
> > >
> > > **Patched Diffusion Model [1]**
> > >
> > > **We have included Patched Diffusion Model [1] in Table 2 (https://pdfhost.io/v/5Gh3jLth8G_table_2, third row).**
> > >
> > > Patched Diffusion Model is a reconstruction-based method (see https://github.com/FinnBehrendt/patched-Diffusion-Models-UAD for illustration). During training, it cuts out patches from images and trains a diffusion model to reconstruct them based on the surrounding context. During inference, it splits an input image into a grid of patches. Diffusion model reconstructs each patch from its noised version based on the remaining clean patches. The reconstructed patches are aggregated into a full image reconstruction, and anomaly scores are obtained as pixel-wise reconstruction errors. Note that, if training dataset contains pathologies, the diffusion model can learn to reconstruct them as well as healthy regions, resulting in False Negative errors. Indeed, we empirically observe this behaviour (​​**see qualitative results at https://pdfhost.io/v/gGueNFpgUv_patched_diffusion_model**).
> > >
> > > *Why did not we implement Masked Diffusion Model [2]?* Initially, we planned to include [2] (https://arxiv.org/abs/2305.19867). However, its official implementation (https://github.com/hasan1292/mDDPM) includes critical pipeline components not described in the paper. Upon closer inspection, we observed that [2] heavily relies on [1] — its Masking Block is applied alongside [1]’s Cut Out during training, and it adopts the same patch-wise pipeline as [1] during inference. Given that these key aspects are not described in [2], we prioritized experiments with [1], as it offers a clearer alignment between the paper and the code.
> > >
> > > **Simplenet [3]**
> > >
> > > **We include experiments with Simplenet [3] in Table 3 (https://pdfhost.io/v/GBHyByNHS4_table_3)**, as it can be used as an alternative to the gaussian and flow-based density models in our framework.
> > >
> > > The main idea of Simplenet is to train a discriminator $d$ (MLP net) to distinguish between descriptors $y$ and their noisy counterparts $y^{\mathrm{noisy}} = y + \varepsilon$, $\varepsilon \sim \mathcal{N}(0, I)$. [3] also uses a so-called adaptor $a$ — a fully-connected layer applied to the descriptors as a trainable pre-processing step before adding noise and applying discriminator. Both adaptor and discriminator are trained to minimize the following objective:
> > >
> > > $$
> > > \mathbb{E}_{y, \varepsilon}[\max(\alpha + d(a(y)), 0) + \max(\alpha - d(a(y) + \varepsilon), 0)] \to \min, \quad (1)
> > > $$
> > >
> > > i.e. enforce $d(a(y))$ to be less than $-\alpha$ and $d(a(y) + \varepsilon)$ to be greater than $\alpha$ for some margin $\alpha > 0$. At the inference stage, discriminator’s pixel-wise predictions $d(a(y))$ are used as anomaly scores.
> > >
> > > The original Simplenet yielded poor results in our experiments: the training loss quickly decreased almost to zero, while validation AUROC remained about 0.5 (https://pdfhost.io/v/xh488GHMZA_simplenet). The reason was that the adaptor simplified the task for the discriminator and the latter did not learn to differentiate between normal and abnormal descriptors. Therefore, we omitted the adaptor and validation AUROC increased up to 0.85 (https://pdfhost.io/v/xh488GHMZA_simplenet). However, anomaly score maps looked overconfident (https://pdfhost.io/v/cRWURz5RXd_simplenet_anomaly_maps). We thought that the original training objective $(1)$ was too restrictive and decided to replace it with standard binary cross-entropy loss (BCE):
> > >
> > > $$
> > > \mathbb{E}_{y, \varepsilon}[-\log \frac{\exp(d(y + \varepsilon))}{\exp(d(y + \varepsilon)) + \exp(d(y))}] \to \min
> > > $$
> > >
> > > With this objective Simplenet achieves validation AUROC 0.9 ((https://pdfhost.io/v/xh488GHMZA_simplenet) and produces continuous anomaly maps (https://pdfhost.io/v/cRWURz5RXd_simplenet_anomaly_maps).
> > >
> > > We also trained conditional Simplenets, by feeding different conditioning variables to the discriminator as additional input. We provide results for both unconditional and conditional Simplenet models with BCE objectives in Table 3 (https://pdfhost.io/v/GBHyByNHS4_table_3). Their results are inferior to those of the flow-based models, probably because the latter explicitly estimate descriptors’ density which can be more appropriate for anomaly scoring.
> > >
> > > **References**
> > >
> > > [1] Behrendt, Finn, et al. "Patched diffusion models for unsupervised anomaly detection in brain mri." Medical Imaging with Deep Learning. PMLR, 2024.
> > >
> > > [2] Iqbal, Hasan, et al. "Unsupervised anomaly detection in medical images using masked diffusion model." International Workshop on Machine Learning in Medical Imaging. Cham: Springer Nature Switzerland, 2023.
> > >
> > > [3] Liu, Zhikang, et al. "Simplenet: A simple network for image anomaly detection and localization." Proceedings of the IEEE/CVF conference on computer vision and pattern recognition. 2023.

---

### Official Review · Reviewer_pobP · 2025-03-14

**Overall Recommendation:** 2

**Summary:**

The paper presents Screener, a framework based on unsupervised visual anomaly segmentation (UVAS) for 3D medical scans. The proposed model aims to reduce the dependency on ground truth (GT) annotations. It is trained on a large dataset of 30K CT scans and evaluated on 1.8K scans, covering a variety of pathological segmentations. The authors claim that the model effectively addresses the challenges of anomaly segmentation without requiring manual annotations.

**Claims And Evidence:**

The authors claim that their method is designed for 3D medical image segmentation. However, this claim appears overstated based on the current evaluation. The framework has only been tested on CT scans, without validation on other key modalities such as MRI. Furthermore, while the authors repeatedly reference segmentation, the model has not been explicitly tested on segmentation tasks, nor are segmentation-specific metrics reported. This omission weakens the claim that the method is applicable to segmentation.

**Essential References Not Discussed:**

The paper is missing references to key related works that are essential for contextualizing its contributions. For instance, the following papers could be discussed:

VISA-FSS: A volume-informed self-supervised approach for few-shot 3D segmentation, MICCAI 2023.
Transformer-based models for unsupervised anomaly segmentation in brain MR images, MICCAI Workshop 2022.

These studies provide valuable insights into self-supervised learning for medical image segmentation and anomaly detection, which are directly relevant to the proposed method.

**Experimental Designs Or Analyses:**

While the experiments are generally well-executed, there are critical gaps in evaluation:

1- Lack of Segmentation Task Evaluation: Given the frequent references to segmentation, the paper should explicitly evaluate segmentation performance and report relevant metrics (e.g., Dice score, IoU).

2- Need for More Comprehensive Analysis: Additional metrics and statistical tests should be included to validate the significance of the results.

3- Lack of Robustness Testing: The method is tested on high-quality CT scans, but its performance on noisy or degraded scans remains unclear. Evaluating robustness to noise, artifacts, and variations in acquisition settings would strengthen the study.

**Methods And Evaluation Criteria:**

The proposed evaluation methodology is incomplete. To substantiate the generalizability of the approach, the method should be evaluated on additional imaging modalities, such as MRI, to ensure its robustness across different medical datasets. Furthermore, the segmentation task itself is not explicitly evaluated, which is necessary given the claims in the title and introduction.
Additionally, Table 2 lacks a sufficient number of comparative baselines, limiting the ability to assess the proposed method's relative performance.

**Other Comments Or Suggestions:**

- Expand the Literature Review: The authors should discuss existing methods that incorporate registration-based approaches for segmentation, as well as the strengths and weaknesses of UVAS compared to other self-supervised pretraining techniques.

- Clarify Key Claims: The paper should explicitly differentiate between anomaly detection and segmentation to avoid overstating its contributions.

**Other Strengths And Weaknesses:**

The paper tackles an important problem and introduces an interesting approach. However, several issues need to be addressed:

1- Limited Scope of Evaluation: The model is tested exclusively on CT scans, and there is no exploration of other medical imaging modalities (e.g., MRI).

2- Potential Bias in Dataset: The qualitative results (Fig. 1) suggest that the scans used are high-quality, but robustness to noisy or low-quality scans is not evaluated.

3- Lack of Comparison with Fully Supervised Methods: The method should be compared with fully supervised segmentation models (e.g., UNet) to assess its performance in a more practical clinical setting.

**Questions For Authors:**

1- How does the proposed method generalize to other medical imaging modalities, such as MRI?

2- Since segmentation is frequently mentioned in the paper, why is segmentation performance not explicitly evaluated with standard metrics?

3- How does the proposed model compare to fully supervised segmentation methods, such as UNet or nnU-Net?

4- What is the method’s robustness to low-quality scans, noise, and artifacts?

**Relation To Broader Scientific Literature:**

The paper builds upon prior work in visual anomaly segmentation (UVAS) for 3D CT scans. The authors review density-based approaches and propose leveraging dense self-supervised learning (SSL) techniques to pre-train feature maps, which are then used in a density-based UVAS framework. This approach is well motivated and aligns with recent trends in self-supervised representation learning for medical imaging.

**Theoretical Claims:**

I reviewed the theoretical claims presented in the paper, and they appear to be correct.

---

> ### Author Rebuttal · Authors · 2025-04-01
>
> Dear Reviewer *pobP*, thank you for your thoughtful review of our submission and for the time you dedicated to providing such detailed and relevant feedback. Your critical comments on our experimental design were particularly valuable, and we have done our best to address them thoroughly in the responses below.
>
> **Inclusion of segmentation metrics**
>
> To provide more relevant and interpretable metrics **we have updated Table 2 (https://pdfhost.io/v/n7Bt5c7zJE_table_2) and Tables 3, 4 (https://pdfhost.io/v/2RLGCYvMgy_tables_3_4) to include Dice scores** and voxel-level AUROCs. AUROC / AUPRO up to 0.3 FPR metrics have been moved to the Appendix.
>
> Initially, we omitted Dice scores due to a mismatch between our problem statement (segmenting all pathologies) and the available ground truth masks (limited to specific pathologies — lung cancer in LIDC, pneumonia in MIDRC, liver tumors in LiTS, kidney tumors in KiTS). Note that this discrepancy leads to an **underestimation** of unsupervised models’ Dice scores, as many test images indeed contain additional anomalies detected by our model but not annotated in the ground truth masks (see Figure 2 and more examples at https://pdfhost.io/v/PNygYvbsYn_mismatch which we will add in the Appendix).
>
> **Comparison with supervised segmentation**
>
> **We have added comparison with a Supervised UNet (Table 2, https://pdfhost.io/v/n7Bt5c7zJE_table_2)**, trained via cross validation on each dataset (LIDC, MIDRC, LiTS and KiTS). Since the supervised model segments only the target pathology, its metrics are naturally higher than those of unsupervised models.
>
> For a fair comparison, **we distilled Screener into UNet and fine-tuned it in a supervised manner (Fine-tuned Screener)**. Namely, at the distillation step, we pre-train UNet (without final sigmoid activation) to predict Screener’s anomaly score maps (using simple MSE loss). Then, at the supervised fine-tuning stage, we randomly re-initialize the pre-trained UNet’s last conv layer and train it on each dataset similarly to Supervised UNet. **Table 2 (https://pdfhost.io/v/n7Bt5c7zJE_table_2) shows that Fine-tuned Screener consistently outperforms Supervised UNet**.
>
> To further demonstrate the significance of the latter result we also have drawn plots (https://pdfhost.io/v/yeejsw47bF_train_sizes) comparing Supervised UNet and Fine-tuned screener trained on datasets’ subsamples of different sizes (10, 20 or 40 images). Note that **when training on 20 images with annotated lung cancer, Fine-tuned Screener achieves 2 times higher Dice scores than Supervised UNet**.
>
> **Robustness testing**
>
> We evaluated Screener on LIDC subsets with varying acquisition settings:
> - **Low-dose (<200 mA) vs. high-dose CTs:** Dice score 0.04±0.12 (low-dose) vs. 0.06±0.13 (high-dose), AUROC 0.94 (low-dose) vs. 0.97 (high-dose) (see ROC curves at https://pdfhost.io/v/8AzVC4JVTU_high_vs_low)
> - **With vs. without contrast agent:** Dice score 0.05±0.13 (with contrast) vs. 0.04±0.13 (without contrast), AUROC 0.96 (with contrast) vs. 0.96 (without contrast) (see ROC curves at https://pdfhost.io/v/8AW5PJr5kQ_contrast_vs_noncontrast)
>
> These results suggest **slightly better performance on high-dose CTs** and **robustness to contrast agents**. However, scatter plots (https://pdfhost.io/v/HLNkq7pd7W_doses) reveal minimal dependence on dose levels. Additionally, we provide examples of Screener’s anomaly maps for both a low-dose scan and an **image with artifacts** (https://pdfhost.io/v/3fBJ42LqH4_robustness). This analysis will be included in the Appendix.
>
> **Experiments on MRI**
>
> While our methodology is theoretically applicable to MRI images, empirical validation would require obtaining official access to MRI datasets and time-consuming experiments with our model and all the baselines. Unfortunately, we will not manage to accomplish this during the rebuttal period.
>
> Given our focus on CT images (one of the project goals was to retrieve images with different abnormalities from our large scale in-house CT database), we propose renaming the paper to: *“Screener: Self-supervised Pathology Segmentation Model for Medical CT Images”* — pending your approval.
>
> **Missing references**
>
> We will discuss the suggested related works (VISA-FSS, Transformer-based models for unsupervised anomaly segmentation) in Section 5.
>
> We hope these revisions address your main concerns. Please let us know if further clarifications are needed to reconsider the score.

---

### Decision · Program_Chairs · 2025-05-01

**Decision:**

Reject

**Comment:**

The paper obtains mixed reviewing comments from four experts.

Reviewer 'pobP' is negative, recommending 'Weak Reject' after rebuttal. Reviewers 'X8PP', 'hWnx' and 'pFWf' are positive, recommending 'Accept' or  'Weak Accept' after rebuttal.

However, the AC believes that the rebuttal has successfully addressed the most concerns of Reviewer 'pobP' (such as the MRI experiment) and hence decides to downweight his/her comments.

So, it is recommend to accep the paper.